# Estimating impact of food choices on life expectancy: A modeling study

Lars T. Fadnes[1,2]*, Jan-Magnus Økland[1,3], Øystein A. Haaland[1,3©], Kjell Arne Johansson[1,2,3©]

1 Department of Global Public Health and Primary Care, University of Bergen, Norway, 2 Bergen Addiction Research, Department of Addiction Medicine, Haukeland University Hospital, Bergen, Norway, 3 Bergen Center for Ethics and Priority Setting, University of Bergen, Norway

© These authors contributed equally to this work.
* lars.fadnes@uib.no

## Abstract

### Background

Interpreting and utilizing the findings of nutritional research can be challenging to clinicians, policy makers, and even researchers. To make better decisions about diet, innovative methods that integrate best evidence are needed. We have developed a decision support model that predicts how dietary choices affect life expectancy (LE).

### Methods and findings

Based on meta-analyses and data from the Global Burden of Disease study (2019), we used life table methodology to estimate how LE changes with sustained changes in the intake of fruits, vegetables, whole grains, refined grains, nuts, legumes, fish, eggs, milk/dairy, red meat, processed meat, and sugar-sweetened beverages. We present estimates (with 95% uncertainty intervals [95% UIs]) for an optimized diet and a feasibility approach diet. An optimal diet had substantially higher intake than a typical diet of whole grains, legumes, fish, fruits, vegetables, and included a handful of nuts, while reducing red and processed meats, sugar-sweetened beverages, and refined grains. A feasibility approach diet was a midpoint between an optimal and a typical Western diet. A sustained change from a typical Western diet to the optimal diet from age 20 years would increase LE by more than a decade for women from the United States (10.7 [95% UI 8.4 to 12.3] years) and men (13.0 [95% UI 9.4 to 14.3] years). The largest gains would be made by eating more legumes (females: 2.2 [95% UI 1.1 to 3.4]; males: 2.5 [95% UI 1.1 to 3.9]), whole grains (females: 2.0 [95% UI 1.3 to 2.7]; males: 2.3 [95% UI 1.6 to 3.0]), and nuts (females: 1.7 [95% UI 1.5 to 2.0]; males: 2.0 [95% UI 1.7 to 2.3]), and less red meat (females: 1.6 [95% UI 1.5 to 1.8]; males: 1.9 [95% UI 1.7 to 2.1]) and processed meat (females: 1.6 [95% UI 1.5 to 1.8]; males: 1.9 [95% UI 1.7 to 2.1]). Changing from a typical diet to the optimized diet at age 60 years would increase LE by 8.0 (95% UI 6.2 to 9.3) years for women and 8.8 (95% UI 6.8 to 10.0) years for men, and 80-year-olds would gain 3.4 years (95% UI females: 2.6 to 3.8/ males: 2.7 to 3.9). Change from typical to feasibility approach diet would increase LE by 6.2 (95% UI 3.5 to 8.1) years for 20-year-old women from the United States and 7.3 (95% UI 4.7

**Data Availability Statement:** All relevant data are within the manuscript and its Supporting information files.

**Funding:** The authors received no specific funding for this work.

**Competing interests:** The authors have declared that no competing interests exist.

**Abbreviations:** 95% UI, 95% uncertainty interval; FA, feasibility approach diet; GBD, Global Burden of Disease study; HR$_a$, alternative hazard ratio; LE, life expectancy; OD, optimized diet; TW, typical Western diet.

to 9.5) years for men. Using NutriGrade, the overall quality of evidence was assessed as moderate. The methodology provides population estimates under given assumptions and is not meant as individualized forecasting, with study limitations that include uncertainty for time to achieve full effects, the effect of eggs, white meat, and oils, individual variation in protective and risk factors, uncertainties for future development of medical treatments; and changes in lifestyle.

## Conclusions

A sustained dietary change may give substantial health gains for people of all ages both for optimized and feasible changes. Gains are predicted to be larger the earlier the dietary changes are initiated in life. The Food4HealthyLife calculator that we provide online could be useful for clinicians, policy makers, and laypeople to understand the health impact of dietary choices.

### Author summary

#### Why was this study done?

- Food is fundamental for health, and globally dietary risk factors are estimated to cause 11 million deaths and 255 million disability-adjusted life years annually.

- The Global Burden of Diseases, Injuries, and Risk Factors study (GBD) provides summary measures of population health that are relevant when comparing health systems but does not estimate the impact of alterations in food group composition and respective health benefits.

- The EAT–Lancet commission did present a planetary diet, but it gives limited information on the health impact of other diets, and few people are able to adhere to strict health maximization approaches.

#### What did the researchers do and find?

- Our modeling methodology using meta-analyses, data from the Global Burden of Disease study and life table methodology showed that life expectancy (LE) gains for prolonged changes from typical Western to optimizing diets could translate into more than a decade for young adults.

- The largest gains would be made by eating more legumes, whole grains and nuts, and less red and processed meat.

- For older people, the gains would be smaller but substantial. Even the feasibility approach diet indicates increased LE by 7% or more for both sexes across age groups.

**What do these findings mean?**

- The online Food4HealthyLife calculator (https://food4healthylife.org/) enables the instant estimation of the effect on LE of a range of dietary changes.

- Understanding the relative health potential of different food groups could enable people to make feasible and significant health gains.

- The Food4HealthyLife calculator could be a useful tool for clinicians, policy makers, and laypeople to understand the health impact of dietary choices.

## Introduction

Food is fundamental for health. Globally, dietary risk factors are estimated to cause 11 million deaths and 255 million disability-adjusted life years annually [1]. Still, navigating within the nutritional research field can be overwhelming to clinicians, policy makers, and even researchers. Since 2017, about 250,000 scientific articles on nutritionally related topics have been published (S1 Text). Fortunately, several recent meta-analyses have summarized the impact on the risk of premature deaths for various food groups, including fruits, vegetables, whole grains and refined grains, nuts and legumes, fish, eggs, milk/dairy, red and processed meats, and sugar-sweetened beverages [2–6].

The Global Burden of Disease study (GBD) provides summary measures of population health that are relevant when comparing health systems [7]. GBD includes population-level estimates for life years lost due to some dietary risk factors [8], but such aggregated health metrics have little relevance when making individual decisions. The EAT–Lancet commission did present a planetary diet that presented a diet balancing health and environmental perspectives [9], but it gives limited information on the health impact of other diets, and few people are able to adhere to strict health maximization approaches [10]. Although the planetary diet and GBD risk factor estimates indicate directions of changes in food intake that are useful, more comprehensive models estimating the impact of various dietary choices on lifetime health are needed.

To better understand the impact on health of dietary choices, we have developed methodology that integrates and presents current knowledge. The availability of such methodology is essential in order to make informed dietary choices at all levels from individuals to policy makers [11]. In this paper, we present new methodology that allows for the estimation of how different diets affect sex- and age-specific life expectancy (LE).

## Methods

The LE at a certain age is the number of years an individual at that age is expected to live before they die given a set of age-specific mortality rates. We used mortality rates extracted from GBD 2019 (published in 2020) [12]. Johansson and colleagues presented a framework for measuring LE from disease onset for specific conditions [13]. We modified this approach by considering "change in diet" as a condition that may have both a positive and a negative health impact. Conceptually, our approach can be summed up as follows:

1. Let $LE_{age}(D)$ be the age-specific LE with prolonged change to diet D.

2. $LE_{age}(D)$ is calculated using standard age-specific lifetable methodology, where annual mortality rates are adjusted according to the selected diet (i.e., mortality rates after the age when the diet is changed are multiplied with the hazard rate corresponding to the change). The baseline diet yields $LE_{age}(D_0)$.

3. Life years gained (or lost) because of change from the baseline diet to diet D is now $LE_{age}(D)$–$LE_{age}(D_0)$.

A more detailed description of the methodology to estimate background LE is given in Johansson and colleagues' paper [13].

Recent meta-analyses provided dose–response data on the impact of various food groups on mortality for the following food groups: whole grains, fruits, vegetables, nuts, legumes, fish, eggs, milk/dairy, refined grains, red meat, processed meat, and sugar-sweetened beverages [2–5]. To identify meta-analyses on these food groups, a search in PubMed dated 26 April 2021 was screened and data extracted from these (see search string in S2 Text). When several meta-analyses were available, we opted for the most comprehensive (usually the latest meta-analyses) with dose–response relationship data unless later less comprehensive meta-analyses argued well for excluding studies. For white meat, we did not have a complete dose–response curve, but a meta-analysis has suggested that the effect on mortality is neutral [14], which also was the case for small amounts of added oils [15]. Most of the studies were adjusted for intake of other food groups and factors such as smoking, exercise, body mass index, age, and sex. Each of the food groups were considered as individual protective or risk factors.

Diets vary between individuals and settings, but as the baseline in our model, we used a "typical Western diet" (TW) based on consumption data from the United States and Europe (S3 Text). The optimized diet (OD) values were set where dose–response data on consumption indicated no additional mortality gain in further increasing or decreasing intake (i.e., the impact on mortality plateaued). As a compromise between the TW and the optimal diet, we also considered a feasibility approach diet (FA), which was chosen as the midpoint for each food group between the typical diet and OD.

In each case, dietary intake was improved from the TW through feasible to optimal levels (rounded off):

- Whole grains (fresh weight): TW 50 g, FA 137.5 g, and OD 225 g (e.g., 2 thin slices of rye bread and 1 small bowl of whole grain cereal, and some whole grain rice). For whole grains, 225 g of fresh weight corresponds to about 75 g dry weight, equivalent of 7 servings/day);

- Vegetables: TW 250 g, FA 325 g, and OD 400 g (5 servings, e.g., 1 big tomato, 1 sweet pepper, mixed salad leaves, a half avocado, and a small bowl of vegetable soup);

- Fruits: TW 200 g, 300 g, and OD 400 g (5 servings, e.g., 1 apple, banana, orange, kiwi, and a handful of berries);

- Nuts: TW 0 g, FA 12.5 g, and OD 25 g (1 handful of nuts);

- Legumes: TW 0 g, FA 100 g, and OD 200 g (e.g., 1 big cup of soaked beans/lentils/peas);

- Fish: TW 50 g, FA 125 g, and OD 200 g (e.g., 1 big slice of herring);

- Eggs: TW 50 g, FA 37.5 g, and OD 25 g (half an egg);

- Milk/dairy: TW 300 g, FA 250 g, and OD 200 g (e.g., 1 cup of yoghurt);

- Refined grains: TW 150 g, FA 100 g, OD 50 g (e.g., refined grains in bread if mixed whole/ refined bread);

- Red meat: TW 100 g, FA 50 g, and OD 0 g;

- Processed meat: TW 50 g, FA 25 g, and OD 0 g;

- White meat: TW 75 g, FA 62.5 g, and OD 50 g;

- Sugar-sweetened beverages: TW 500 g, FA 250 g, and OD 0 g;

- Added plant oils: TW 25 g, FA 25 g, and OD 25 g.

Other food groups were not considered. To avoid reporting estimates for insufficiently studied and unsustainable diet alternatives, the model does not report estimates if the total energy consumption for the diet input was below 4,000 kJ/day or above 16,000 kJ/day. Energy estimates per food group were obtained from a food content database [16]. The energy estimates were 8,085 kJ/day for TW, 7,850 kJ/day for FA, and 7,615 kJ/day for OD. The effect of energy restriction on longevity was not considered.

Health gains from diet changes are generally linked to reduction in cardiovascular disease, cancer, and diabetes mortality [2–5], all among the leading causes of mortality globally [17]. It has earlier been assumed that reversing the process of cardiovascular disease following reductions in major cardiovascular risk factors would require decades, but it has later been argued that cardiovascular disease mortality can change more quickly within a few years [18,19]. For cancers, the time perspective is likely to be longer. It has been indicated for associations between fruit and vegetable consumption and risk of lung cancer that associations for studies with more than 10 years of follow-up on fruits and vegetables are stronger than those with less than 10 years [20]. More evidence on the time perspective is available for risk factors such as tobacco, where meta-analyses for duration of smoking has indicated that associations between duration of tobacco smoking and risk of lung cancer is substantially higher with 50 years of smoking than 20 years of smoking [21]. To balance between the time perspectives related to both cardiovascular disease and cancer while weighting in the morbidity burden, we assumed that time to full effect was 10 years with a gradual, linear increase in effect (e.g., the effect was 20% of maximum after 2 years). We also conducted sensitivity analyses with 5 years, 30 years, and 50 years to full effect.

We used the following approach to calculate 95% uncertainty intervals (95% UIs) for the overall and food specific effect on LE of dietary changes: First, we extracted confidence intervals for the hazard rates for the proposed changes in intake of each food group from meta-analyses. Then, using a uniform distribution, we drew a number between the upper and lower 95% confidence interval for each food group and used this as input in the model. This procedure was repeated 200 times (with a fixed seed as starting point), and 95% uncertainty limits were selected as the 2.5 and 97.5 percentiles. Even though most meta-analyses adjusted for intake of other food groups, there is a possibility of different food groups presenting overlapping gains and thus overestimating the effects of each food group. Conversely, it is also possible that meta-analyses have overadjusted estimates so that the hazard ratios are closer to the null than the true effects. To take these effects into account, we conducted a new set of sensitivity analyses. In these analyses, we calculated alternative hazard ratios ($HR_a$) based on $HR_0$, the hazard ratio from the meta-analyses for a given change of intake for a given food group. Assuming first that $HR_0 < 1$, we use the formula

$$HR_a = HR_0 + (1 - HR_0) * (1 - m),$$

where m is a parameter taking on values from 0.5 to 1.5. If $0.5 < m < 1$, the model becomes more conservative in the sense that the effect of dietary changes is reduced, whereas if $1 < m \le 1.5$, the model becomes more "radical," in that the effect is amplified. When $HR_0 > 1$,

we use $HR_0^* = 1 / HR_0$ in (1) to get

$$HR_a* = HR_0* + (1 - HR_0*) \times (1 - m),$$

and then finally $HR_a = 1 / HR_a^*$.

In addition to the 95% UIs, we report sensitivity adjusted uncertainty intervals where the central estimate of the model is based on $HR_0$ (i.e., m = 1), the lower interval is when m = 0.5 as and similarly the upper interval when m = 1.5.

Data on background mortality from 2019 for specific countries and regions were obtained from the freely available GBD cause of death database [12]. We extracted data for the United States, China, and Europe, as these are the regions from where most of the nutritional studies providing mortality estimates originate. Region-specific estimates on total mortality rates in 5-year age groups were also available from GBD. These were converted to single-year age-specific mortality rates in our model.

To assess the quality of evidence for each food group from the meta-analyses, we use Nutri-Grade, a version of GRADE adapted to nutritional studies [22]. Certainty of evidence is categorized as "very low" (0 to 3.99), "low" (4 to 5.99), "moderate" (6 to 7.99), or "high" (8 to 10). The quality of evidence was "high" for whole grains (NutriGrade score: 8), "moderate" for fish (7.75), processed meat (7.5), nuts (7), red meat (6.5), legumes (6), and dairy (6), "low" for vegetables (5.8), fruits (5.8), SSBs (5.5), and refined grains (5), and "very low" for eggs (3.8) and white meat (2). We further constructed an overall quality score by taking the mean of the NutriGrade scores for each of the food groups weighted by their absolute contribution to LE. The quality of the meta-analyses was assessed with the AMSTAR–2 tool [23]. The quality of the meta-analyses was rated as high for studies on all included meta-analyses [2–5,15], except for the meta-analysis on white meat that was rated as moderate [14].

We used the R package Shiny to create a web application (https://food4healthylife.org/) that enables the estimation of the effect of a range of dietary changes (S1 Fig). In the left food panel (i.e., the diet before change), the defaults are set to the "typical diet." The right food panel represents diet after change. Clicking the "*Optimal*" or "*Feasible*" button, the right panel of sliders are adjusted to the 2 OD and FA diet patterns. In this paper, we present estimated gain in LE when changing from a typical diet to OD or FA for 20-, 40-, 60-, and 80-year-old adults from the United States, China, and Europe. Graphs including forest plots are calculated in Stata SE 17.0 (including the *admetan* package).

Only publicly available data sources have been used, and thus no ethical permission is required. We adhered to the transparent reporting of a multivariable prediction model for individual prognosis or diagnosis (TRIPOD; see S1 TRIPOD Checklist) [24].

## Results

In this section, we will focus on the United States, but the results for China and Europe were generally very similar (can be found in S2–S15 Figs). Table 1 and Fig 1 estimate the life expectancies at different ages associated with a typical Western diet, a feasibility approach diet, and an optimized diet. As seen, an increase in LE of up to 13.0 years (95% UI 9.4 to 14.3) is possible for male 20-year-olds from the United States by sustained dietary changes, and even for 80-year-olds, gains of 3.4 years (95% UI 2.7 to 3.9) are possible. Corresponding numbers for 20- and 80-year-old females are 10.7 years (95% UI 8.4 to 12.3) and 3.4 years (95% UI 2.6 to 3.8). Still, prolonged dietary changes at age 20 years would give about 48% higher gain in LE as changes starting from age 60 years, and 3 times the gains when compared with changes starting at age 80 years (Figs 2 and 3). Similar findings were seen for China and the United States.

**Table 1. LE for males and females at different ages from the United States, China, and Europe for different diets.** Gain in LE when changing from a typical Western diet to a feasibility approach or optimized diet is also indicated.

| Region | | Typical Western | | Feasibility approach | | | | Optimized | | | |
|---|---|---|---|---|---|---|---|---|---|---|---|
| | | Male | Female | Male | | Female | | Male | | Female | |
| | Age | LE | LE | LE | Gain | LE | Gain | LE | Gain | LE | Gain |
| United States | 20 | 57.8 | 62.5 | 65.1 | 7.3 | 68.7 | 6.2 | 70.8 | 13.0 | 73.3 | 10.7 |
| | 40 | 39.4 | 43.3 | 46.0 | 6.5 | 49.0 | 5.7 | 51.1 | 11.7 | 53.3 | 10.0 |
| | 60 | 22.4 | 25.3 | 27.2 | 4.8 | 29.9 | 4.5 | 31.2 | 8.8 | 33.3 | 8.0 |
| | 80 | 9.0 | 10.3 | 10.9 | 1.9 | 12.3 | 2.0 | 12.4 | 3.4 | 13.7 | 3.4 |
| China | 20 | 56.7 | 61.8 | 63.7 | 7.0 | 67.7 | 5.9 | 69.6 | 12.9 | 72.4 | 10.6 |
| | 40 | 37.6 | 42.2 | 44.1 | 6.4 | 47.8 | 5.6 | 49.7 | 12.0 | 52.4 | 10.2 |
| | 60 | 20.1 | 23.5 | 25.0 | 4.9 | 28.2 | 4.7 | 29.4 | 9.3 | 32.1 | 8.6 |
| | 80 | 7.4 | 8.6 | 9.1 | 1.7 | 10.5 | 1.9 | 10.5 | 3.1 | 12.0 | 3.4 |
| Europe | 20 | 56.3 | 62.9 | 63.8 | 7.6 | 68.8 | 5.9 | 69.9 | 13.7 | 73.3 | 10.4 |
| | 40 | 37.7 | 43.4 | 44.5 | 6.8 | 49.0 | 5.5 | 50.0 | 12.3 | 53.2 | 9.8 |
| | 60 | 21.0 | 25.1 | 25.9 | 4.9 | 29.6 | 4.5 | 30.0 | 9.1 | 33.2 | 8.1 |
| | 80 | 8.4 | 9.8 | 10.3 | 1.8 | 11.7 | 2.0 | 11.7 | 3.3 | 13.2 | 3.5 |

*For the optimal diet and feasibility approach diet, the following intakes were used: 225 g and 137.5 g whole grains (fresh weight), 400 g and 325 g vegetables, 400 g and/ 300 g fruits, 25 g and 12.5 g nuts, 200 g and/ 100 g legumes, 200 g and 100 g fish, 25 g and 37.5 g eggs, 200 g and 250 g milk/dairy, 50 g and 100 g refined grains, 0 g and 50 g red meat, 0 g and 25 g processed meat, 50 g and 62.5 g white meat, 0 g and 250 g sugar-sweetened beverages, and 25 g and 25 g added plant oils.

LE, life expectancy.

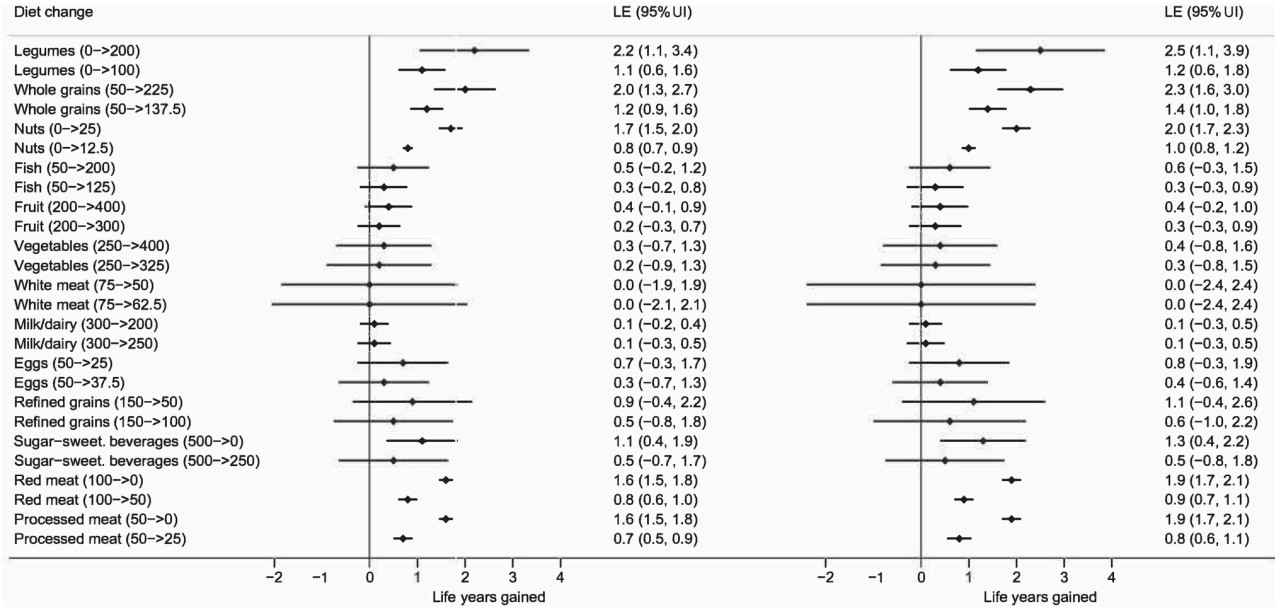

**Fig 1. Expected life years gained for 20-year-old female adults (left forest plot) and males (right forest plot) from the United States who change from a typical Western diet to an optimized or feasible approach diet with changes indicated in grams per day.** Estimates per food groups and total change in LE is presented with uncertainty intervals (UI). *The meta-evidence is high for whole grains; moderate for fish, nuts, legumes, processed and red meat, and sugar-sweetened beverages; and low for and very low for white meat. LE, life expectancy; 95% UI, 95% uncertainty interval.

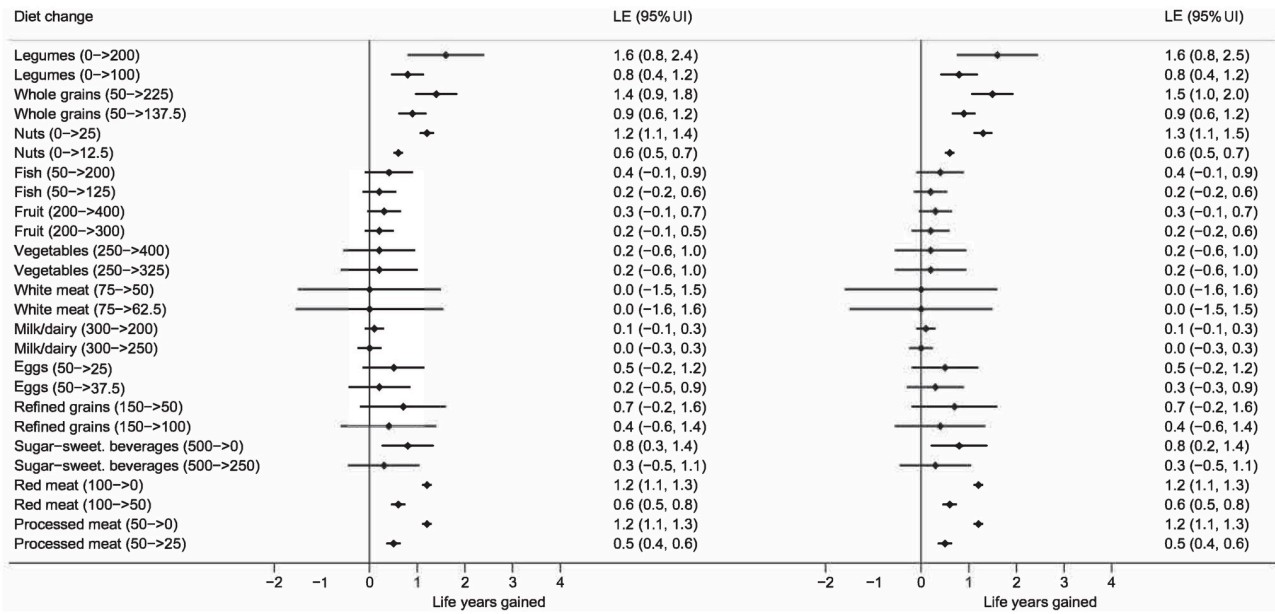

**Fig 2. Expected life years gained for 60-year-old female adults (left forest plot) and males (right forest plot) from the United States who change from a typical Western diet to an optimized or feasible approach diet with changes indicated in grams per day.** Estimates per food groups and total change in LE is presented with uncertainty intervals (UI). *The meta-evidence is high for whole grains; moderate for fish, nuts, legumes, processed and red meat, and sugar-sweetened beverages; and low for and very low for white meat. LE, life expectancy; 95% UI, 95% uncertainty interval.

Changing from a typical diet to the feasibility approach diet would also give substantial gains for all age groups.

When changing from a typical Western to an optimized diet, the largest gains in LE could be made by eating more legumes, whole grains, and nuts, as well as eating less red meat and processed meat, with gradual reduction in effect with increasing age (Fig 2 and S2 Table). For a 20-year-old from the United States, LE would increase by more than 1 year for each of these food groups. Fruits and vegetables as well as fish had substantial positive impact, but the intake in a typical diet is closer to an optimal intake than for legumes, whole grains, and nuts.

S3 Table indicates that when increasing time to full effect from 10 years to 30 years, gains in LE were reduced by less than 1 year for 20-year-olds (i.e., by 4% to 7%), but the gains for 60-year-olds and 80-year-olds were reduced by 35% to 71%. Conversely, decreasing time to full effect from 10 years to 5 years (S16 Fig, S3 Table), health gains for 20-year-old females and males increased by 0.1 to 0.2 years (i.e., by 1% to 2%), whereas gains increased by 0.5 to 0.8 years for 60-year-olds (i.e., 6% to 9%) and 1.2 to 1.3 years for 80-year-olds (i.e., 35% to 38%) (S17 Fig).

The overall quality of evidence was moderate for the optimized diet (NutriGrade score: 6.5) and identical for the feasibility approach diet (NutriGrade score: 6.5).

## Discussion

In this paper, we present a method for estimating the impact of food choices on LE. This method has been implemented in a tool that is freely available online—the Food4HealthyLife calculator. Our results indicate that for individuals with a typical Western diet, sustained dietary changes at any age may give substantial health benefits, although the gains are the largest if changes start early in life.

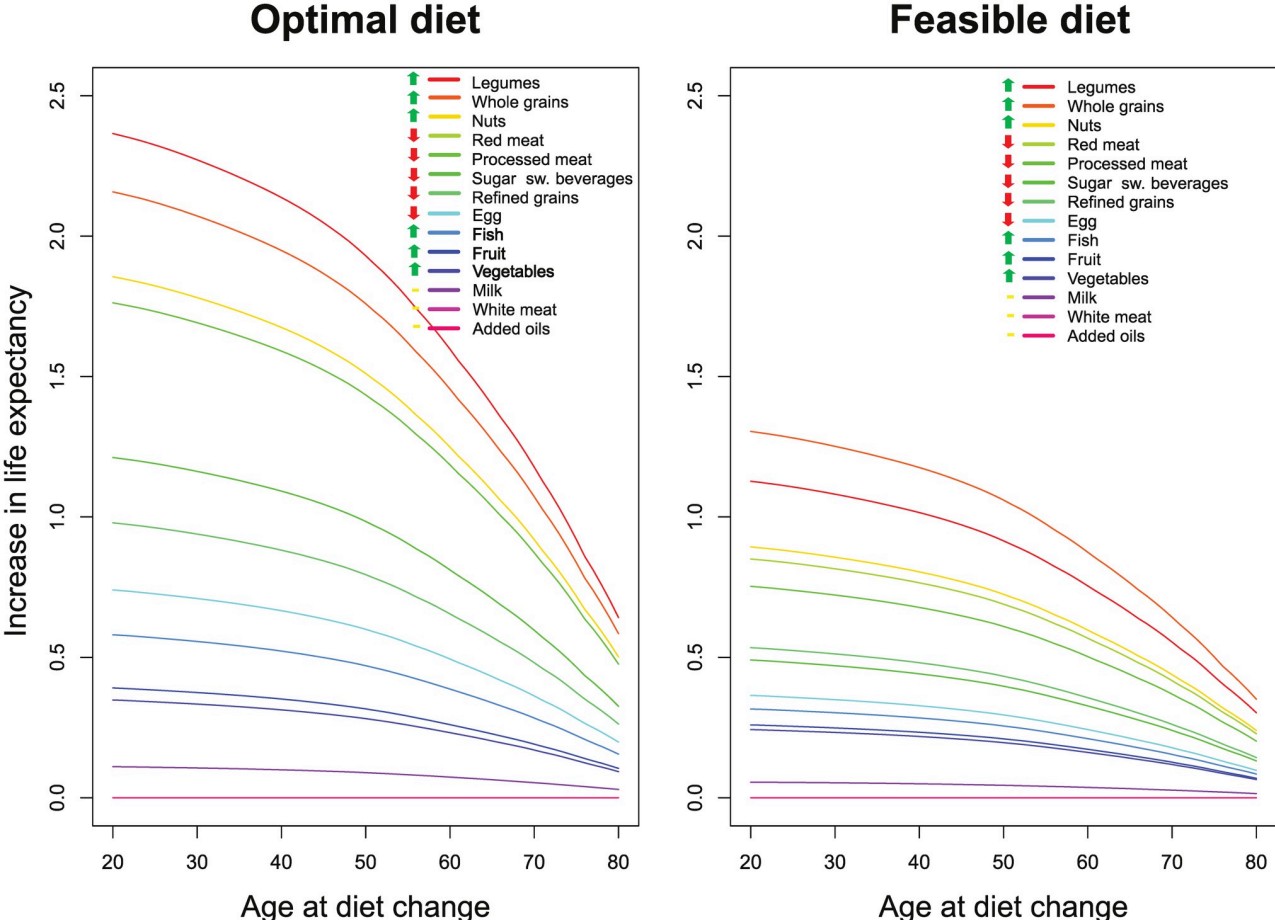

**Fig 3. Expected increase in LE for optimizing different food groups with diet changes initiating from various ages between 20 and 80 years of age (left plot).** Right plot presents similar estimates with a feasible approach* diet (time to full effect: 10 years). *For the optimal diet and feasibility approach diet, the following intakes were used: 225 g and 137.5 g whole grains (fresh weight), 400 g and 325 g vegetables, 400 g and/ 300 g fruits, 25 g and 12.5 g nuts, 200 g and/ 100 g legumes, 200 g and 100 g fish, 25 g and 37.5 g eggs, 200 g and 250 g milk/dairy, 50 g and 100 g refined grains, 0 g and 50 g red meat, 0 g and 25 g processed meat, 50 g and 62.5 g white meat, 0 g and 250 g sugar-sweetened beverages, and 25 g and 25 g added plant oils. Note that lines for LE for red and processed meat changes are overlapping and similarly also for white meat and added oils. LE, life expectancy.

Eating more legumes, whole grains, and nuts, and eating less red meat and processed meats were estimated to be the most effective ways to increase LE for individuals with a typical diet. This reflects a combination of the health effect for each food group combined with the difference between typical and optimal intakes. Meta-analyses have also shown strong positive health effects from fruits, vegetables, and fish [2,5]. However, for these food groups, the typical intake was closer to optimal intake than for other food groups, particularly for vegetables. One could argue that for some food groups such as legumes, an optimal diet requires large intake and that such intakes might be unfeasible for many. Thus, we have also presented feasibility approach diet estimates that are closer to what we may realistically expect from diet changes of most people in most settings where ideals often are difficult to follow in practice. However, for most food groups, our estimates in the feasibility approach are within ranges that are common in cohort studies. There are also substantial individual variations in diet profile, which has impact on the potential health gain for each food group. As an example, some people have diets that are relatively similar to optimized diets and can expect less additional benefits from optimizing diets compared to individuals with a typical Western diet. Our food outcome

calculation could take such variations at baseline into account by using different assumptions on nutrition starting points beyond what is presented here as default for a typical "Western diet."

For several of the food groups, more than one meta-analysis is available. For red and processed meats, a more recent meta-analysis from 2019 than the one used in our estimates has been published [6]. However, this did not present dose–response data for red and processed meats separately, and the supplemental data for these groups combined indicated similar results as for the meta-analysis by Schwingshackl and colleagues. It is worthy to note that meta-analyses indicate worse outcomes on LE from processed meat than nonprocessed red meat when compared by weight, but if the consumption of unprocessed red meat consumption is double as high as for processed meat, the total effect is probably similar. For fish, whole grains, and legumes, more recent but smaller and less comprehensive meta-analyses were omitted from our data [25–27]. These also provided similar effect estimates to the estimates we used. For some food groups such as dairy products, fruits, and vegetables, systematic reviews of meta-analyses were available and supported the selection of the data sources [28,29]. For added oils, there were mixed results depending on type of oil, where monounsaturated fatty acids such as olive oil have been reported to have beneficial effects [15,30,31]. As most added oils contain a combination of different types of fatty acids, the general trend for health impact of added oils is often neutral [15]. Many of the background studies were adjusted for other food groups. It can be argued that food groups are interrelated and thus not independent. Studies presenting outcome measures with and without adjustment for other food groups have generally indicated minimal changes in the outcome measures [32–34]. To account for this possibility, we added sensitivity analyses model adjustment.

Our method has several strengths. First, our food impact estimates are from the most comprehensive and recent meta-analyses presenting dose–response data on diet patterns and mortality. We also have developed methodology that integrates different aspects such as time to full effects and potentially some degree of overlapping with sensitivity analyses and uncertainty intervals.

Our method also has several limitations. Meta-analyses present associations and some caution must be used when interpreting these. Still, meta-analyses are in many cases the best available evidence available as trials on diets could be challenging and, in several cases, could be unethical. Thus, emphasized several sensitivity analyses. For some food groups, meta-analyses presenting dose–response data were not available, which yield more uncertainty in model output.

The meta-analyses used in these data had high quality [23], while the meta-evidence ranged from very low (eggs and white meat) to high (whole grains) with most in the moderate quality category [22]. The overall meta-evidence was estimated as moderate for the optimal and feasibility approach diets. Still, the quality of the evidence for diet changes mostly involving eggs and white meat would be lower than when diet changes are dominated by whole grains, fish, processed meat, and nuts. This is reported in the tool for transparency. For added oils, it is likely that olive oils that are rich in monounsaturated fatty acids have beneficial effects and are probably superior to several other added oils [15,30,31]. However, we did not have sufficient data to present different oils separately.

GBD provides background epidemiological data for the populations we have presented but involves a combination of background data and modeling. We have no information on the impact on past morbidity experienced due to disease, and this was therefore not included in the model, although different health profiles may be associated with different impact of food choices. Thus, our estimates are based on population distributions of health indicators and do not account for differences in risk factors nor genetic vulnerability. The time perspective of

diet change adds another layer of uncertainty. The duration of changes in the studies varies, and it is likely that short-term changes yield weaker effects than those presented in this article. We assumed 10 years to achieve full effects while conducting sensitivity analyses for both 5, 30, and 50 years. Still, progress in development of medical treatments and continuous changes in lifestyle can affect the impact of diet on LE and thus add uncertainty to our estimates [35]. Thus, the methodology is not meant as individualized forecasting of life years gained, but rather population estimates under certain assumptions.

Even though the diet approaches were relatively similar in energy, energy differences may have played a role in the relationships presented, and meta-analyses indicate that patterns in line with the optimal diet are likely to reduce the risk of obesity/overweight [36]. From the literature, we also know that one's diet has a large impact on health-related quality of life [2–4,36–39]. Although we do not model nonfatal effects, LE is correlated with healthy life years. Most of the background data are adjusted for factors such as smoking, exercise, age, and sex. However, some residual confounding may still affect the estimates. Further, we have not considered any long-term health consequences that are due to sustained excessive intake of food with high levels of toxins, such as dioxins and polychlorinated biphenyls, which are relevant for some types of fish and sea foods [40,41]. This is more likely to overestimate than underestimate effect sizes. There is also a risk of overadjustment as some of the studies included in meta-analyses adjusted for potential intermediate factors. This may contribute to underestimating the full impact on dietary changes on health. Model development often have iterative improvements that will gradually give more precise estimates; however, the main messages are likely to be robust. Our sensitivity analyses indicate how the estimated changes in LE due to dietary changes vary if the true effects are over- or underestimated. Even the most conservative approaches indicate strong effects.

In conclusion, sustained change from a typical to an optimized diet from early age could translate into an increase in LE of more than 10 years. Gains are reduced substantially with delayed initiation of changes, particularly when approaching the age of 80 years. An increase in the intake of legumes, whole grains, and nuts, and a reduction in the intake of red meat and processed meats, contributed most to these gains. Fruits and vegetables also have a positive health impact, but for these food groups, the intake in a typical Western diet is closer to the optimal intake than for the other food groups. The Food4HealthyLife calculator could be a useful tool for both clinicians, policy makers, and laypeople to understand impact of various food choices.

## Supporting information

**S1 Text. Medline/PubMed search to estimate number of nutritional articles per year.**
(PDF)

**S2 Text. String used in PubMed to identify meta-analyses for setting hazard ratios.**
(PDF)

**S3 Text. Estimated intake of various food groups in the United States and Norway.**
(PDF)

**S1 Table. Hazard ratios (with uncertainty intervals) for various food groups with uncertainty limits (orange/red labels).**
(PDF)

**S2 Table. Increase in LE for each food group change for 20- and 60-year-old female and male adults from the United States, who change from a TW to OD or FA.** FA, feasibility

approach diet; LE, life expectancy; OD, optimized diet; TW, typical Western diet.
(PDF)

**S3 Table. Absolute and relative change in LE with delay to full effects of 10 (default), 5, 30, and 50 years for 20-, 40-, 60-, and 80-year-old females and males from the United States.** LE, life expectancy.
(PDF)

**S1 Fig. Example of calculator input and output.**
(PDF)

**S2 Fig. Expected life years gained for 20-year-old female adults from China who change from a typical Western diet to an optimized or feasible approach diet with changes indicated in grams.**
(PNG)

**S3 Fig. Expected life years gained for 20-year-old male adults from China who change from a typical Western diet to an optimized or feasible approach diet with changes indicated in grams.** Estimates per food group and change in LE are presented with 95% UIs. LE, life expectancy; 95% UI, 95% uncertainty interval.
(PNG)

**S4 Fig. Expected life years gained for 20-year-old female adults from Europe who change from a typical Western diet to an optimized or feasible approach diet with changes indicated in grams.** Estimates per food group and change in LE are presented with 95% UIs. LE, life expectancy; 95% UI, 95% uncertainty interval.
(PNG)

**S5 Fig. Expected life years gained for 20-year-old male adults from Europe who change from a typical Western diet to an optimized or feasible approach diet with changes indicated in grams.** Estimates per food group and change in LE are presented with 95% UIs. LE, life expectancy; 95% UI, 95% uncertainty interval.
(PNG)

**S6 Fig. Expected life years gained for 20-year-old female adults from the United States who change from a typical Western diet to an optimized or feasible approach diet.** Estimates per food group and change in LE are presented with 95% UIs. LE, life expectancy; 95% UI, 95% uncertainty interval.
(PNG)

**S7 Fig. Expected life years gained for 20-year-old male adults from the United States who change from a typical Western diet to an optimized or feasible approach diet.** Estimates per food group and change in LE are presented with 95% UIs. LE, life expectancy; 95% UI, 95% uncertainty interval.
(PNG)

**S8 Fig. Expected life years gained for 40-year-old female adults from the United States who change from a typical Western diet to an optimized or feasible approach diet.** Estimates per food group and change in LE are presented with 95% UIs. LE, life expectancy; 95% UI, 95% uncertainty interval.
(PNG)

**S9 Fig. Expected life years gained for 40-year-old male adults from the United States who change from a typical Western diet to an optimized or feasible approach diet.** Estimates

per food group and change in LE are presented with 95% UIs. LE, life expectancy; 95% UI, 95% uncertainty interval.
(PNG)

**S10 Fig. Expected life years gained for 60-year-old female adults from the United States who change from a typical Western diet to an optimized or feasible approach diet.** Estimates per food group and change in LE are presented with 95% UIs. LE, life expectancy; 95% UI, 95% uncertainty interval.
(PNG)

**S11 Fig. Expected life years gained for 60-year-old male adults from the United States who change from a typical Western diet to an optimized or feasible approach diet.** Estimates per food group and change in LE are presented with 95% UIs. LE, life expectancy; 95% UI, 95% uncertainty interval.
(PNG)

**S12 Fig. Expected life years gained for 80-year-old female adults from the United States who change from a typical Western diet to an optimized or feasible approach diet.** Estimates per food group and change in LE are presented with 95% UIs. LE, life expectancy; 95% UI, 95% uncertainty interval.
(PNG)

**S13 Fig. Expected life years gained for 80-year-old male adults from the United States who change from a typical Western diet to an optimized or feasible approach diet.** Estimates per food group and change in LE are presented with 95% UIs. LE, life expectancy; 95% UI, 95% uncertainty interval.
(PNG)

**S14 Fig. Expected life years gained for 20-, 40-, 60-, and 80-year-old male and female adults from the US, China, and EU, who change from a typical Western diet to an optimized*** **(labeled "Optimal") or a feasibility approach diet*** **(labeled "Feasible").** Estimates for change in LE is presented with 95% UIs. *For the optimal diet and feasibility approach diet, the following intakes were used: 225/137.5 g whole grains (fresh weight), 400/325 g vegetables, 400/300 g fruits, 25/12.5 g nuts, 200/100 g legumes, 200/125 g fish, 25/37.5 g eggs, 200/250 g milk/dairy, 50/100 g refined grains, 0/50 g red meat, 0/25 g processed meat, 50/62.5 g white meat, 0/250 g sugar-sweetened beverages, and 25/25 g added plant oils. **F20 indicates 20-year-old females, and M60 indicates 60-year-old males. Uncertainty intervals for some food groups have rounding differences compared to corresponding S2 Table due to symmetrical adjustment in the admetan package in Stata. EU, Europe; LE, life expectancy; US, United States; 95% UI, 95% uncertainty interval.
(PNG)

**S15 Fig. Expected life years gained for 20-, 40-, 60-, and 80-year-old male and female adults from the US, China, and EU, who change from a typical Western diet to an optimized*** **(labeled "Optimal") or a feasibility approach diet*** **(labeled "Feasible").** Estimates for change in LE is presented with sensitivity adjusted uncertainty intervals using lower interval as model adjustment of 0.5 and upper interval as model adjustment of 1.5. *For the optimal diet and feasibility approach diet, the following intakes were used: 225/137.5 g whole grains (fresh weight), 400/325 g vegetables, 400/300 g fruits, 25/12.5 g nuts, 200/100 g legumes, 200/125 g fish, 25/37.5 g eggs, 200/250 g milk/dairy, 50/100 g refined grains, 0/50 g red meat, 0/25 g processed meat, 50/62.5 g white meat, 0/250 g sugar-sweetened beverages, and 25/25 g added plant oils. **F20 indicates 20-year-old females, and M60 indicates 60-year-old males.

Uncertainty intervals for some food groups have rounding differences compared to corresponding S2 Table due to symmetrical adjustment in the admetan package in Stata. EU, Europe; LE, life expectancy; US, United States.
(PNG)

**S16 Fig. Expected increase in LE for optimizing different food groups with diet changes initiating from various ages between 20 and 80 years of age (time to full effect: 5 years).**
*For the optimal diet and feasibility approach diet, the following intakes were used: 225 g and 137.5 g whole grains (fresh weight), 400 g and 325 g vegetables, 400 g and/ 300 g fruits, 25 g and 12.5 g nuts, 200 g and/ 100 g legumes, 200 g and 100 g fish, 25 g and 37.5 g eggs, 200 g and 250 g milk/dairy, 50 g and 100 g refined grains, 0 g and 50 g red meat, 0 g and 25 g processed meat, 50 g and 62.5 g white meat, 0 g and 250 g sugar-sweetened beverages, and 25 g and 25 g added plant oils. LE, life expectancy.
(PDF)

**S17 Fig. Expected increase in LE for optimizing different food groups with diet changes initiating from various ages between 20 and 80 years of age (time to full effect: 30 years).**
*For the optimal diet and feasibility approach diet, the following intakes were used: 225 g and 137.5 g whole grains (fresh weight), 400 g and 325 g vegetables, 400 g and/ 300 g fruits, 25 g and 12.5 g nuts, 200 g and/ 100 g legumes, 200 g and 100 g fish, 25 g and 37.5 g eggs, 200 g and 250 g milk/dairy, 50 g and 100 g refined grains, 0 g and 50 g red meat, 0 g and 25 g processed meat, 50 g and 62.5 g white meat, 0 g and 250 g sugar-sweetened beverages, and 25 g and 25 g added plant oils. LE, life expectancy.
(PDF)

**S1 TRIPOD Checklist. Checklist for prediction model development.**
(PDF)

## Acknowledgments

Thanks to Arngeir Berge for assistance with images.

## Author Contributions

**Conceptualization:** Lars T. Fadnes, Jan-Magnus Økland, Øystein A. Haaland, Kjell Arne Johansson.

**Data curation:** Lars T. Fadnes, Jan-Magnus Økland.

**Formal analysis:** Lars T. Fadnes, Jan-Magnus Økland, Øystein A. Haaland.

**Funding acquisition:** Kjell Arne Johansson.

**Investigation:** Lars T. Fadnes, Jan-Magnus Økland, Øystein A. Haaland, Kjell Arne Johansson.

**Methodology:** Lars T. Fadnes, Jan-Magnus Økland, Øystein A. Haaland, Kjell Arne Johansson.

**Project administration:** Lars T. Fadnes, Kjell Arne Johansson.

**Resources:** Lars T. Fadnes.

**Software:** Jan-Magnus Økland.

**Supervision:** Lars T. Fadnes, Øystein A. Haaland, Kjell Arne Johansson.

**Validation:** Lars T. Fadnes, Jan-Magnus Økland, Øystein A. Haaland, Kjell Arne Johansson.

**Visualization:** Lars T. Fadnes, Jan-Magnus Økland, Øystein A. Haaland.

**Writing – original draft:** Lars T. Fadnes.

**Writing – review & editing:** Lars T. Fadnes, Jan-Magnus Økland, Øystein A. Haaland, Kjell Arne Johansson.

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
