## [Editor Report · Decision Letter 0]

21 Sep 2021

Dear Dr Fadnes, 

Thank you for submitting your manuscript entitled "Food for a Healthy Life: Estimating Impact of Food Choices on Life Expectancy" for consideration by PLOS Medicine.

Your manuscript has now been evaluated by the PLOS Medicine editorial staff and I am writing to let you know that we would like to send your submission out for external assessment.

However, we first need you to complete your submission by providing the metadata that is required for full assessment. To this end, please login to Editorial Manager where you will find the paper in the 'Submissions Needing Revisions' folder on your homepage. Please click 'Revise Submission' from the Action Links and complete all additional questions in the submission questionnaire.

Please re-submit your manuscript within two working days, i.e. by Sep 23 2021 11:59PM.

Once your full submission is complete, your paper will undergo a series of checks in preparation for full assessment.

Kind regards,

Richard Turner, PhD

rturner@plos.org

---

## [Decision Letter · Decision Letter 1]

20 Oct 2021

Dear Dr. Fadnes,

Thank you very much for submitting your manuscript "Food for a Healthy Life: Estimating Impact of Food Choices on Life Expectancy" (PMEDICINE-D-21-04011R1) for consideration at PLOS Medicine. 

Your paper was discussed among the editors and sent to independent reviewers, including a statistical reviewer. The reviews are appended at the bottom of this email and any accompanying reviewer attachments can be seen via the link below:

[LINK]

In light of these reviews, we will not be able to accept the manuscript for publication in the journal in its current form, but we would like to invite you to submit a revised version that addresses the reviewers' and editors' comments fully. You will appreciate that we cannot make a decision about publication until we have seen the revised manuscript and your response, and we expect to seek re-review by one or more of the reviewers. 

We hope to receive your revised manuscript by Nov 10 2021 11:59PM. Please email us (plosmedicine@plos.org) if you have any questions or concerns.

Please let me know if you have any questions, and we look forward to receiving your revised manuscript. 

Sincerely,

Richard Turner, PhD

Senior editor, PLOS Medicine

rturner@plos.org

Please adapt the title to better match journal style, including a study descriptor, and we suggest: "Estimating impact of food choices on life expectancy: A modeling study".

In the abstract, please add a new final sentence to the "Methods and findings" subsection, which should begin "Study limitations include ..." or similar and should quote 2-3 of the study's main limitations. 

In the abstract and main text, please state the year of the GBD data used. 

We suspect that the statement "The quality of evidence was moderate (NutriGrade)" (abstract) will need a few words of explanation.

Please move the author summary after the abstract.

Please adapt the author summary so that the third subsection is titled "What do these findings mean?"; we recommend about three points in each of the three subsections. 

Noting "... and even lay people" in the author summary, we suggest removing the word "even", which could appear slightly patronising. 

Throughout the text, please move reference call-outs before punctuation (e.g., "... annually [1].").

Please remove information on funding, competing interests and data sharing from the end of the main text. In the event of publication, this information will appear in the article metadata, via entries in the submission form. 

Please move the information on ethics approvals to the Methods section (main text). 

In the reference list, please remove the information on competing interests from reference 18 and any other relevant references. 

Thank you for including the completed TRIPOD checklist. Please rename the attachment "S1_TRIPOD_Checklist" or similar, and refer to it by this label in the Methods section (main text). 

Please adapt the checklist so that individual items are referred to by section (e.g., "Methods") and paragraph number, not by line or page numbers as these generally change in the event of publication. 

Comments from the reviewers:

*** Reviewer #1: 

Estimating Impact of Food Choices on Life Expectancy

Fadnes at al. PLoS Med PMEDICINE-D-21-04011R1 

This is a very interesting study, generally well written, with potentially important policy messages.

The life-table modelling approach used effect sizes from GBD meta-analyses and data to estimate life expectancy (LE) changes with moderate (FA) or large (OD) sustained changes in the intake of fruits, vegetables, whole grains, refined grains, nuts, legumes, fish, eggs, milk/dairy, red meat, processed meat, and sugar-sweetened beverages. 

The paper presents estimated gain in life expectancy when changing from a typical diet to OD or FA for 20-, 40-60- and 80-year-old adults from Europe, China, and the United States.

Changing from a typical western diet to the optimal diet could increase LE by more than a decade for European men & women, if starting at 20, but with still useful gains even if starting at a later age.

Changing to a more feasible but less radicle diet might generate about half the benefit. 

The underlying modelling idea is ambitious, and doubly impressive given the apparent lack of external funding. 

At present, the analysis is handicapped by a couple of major limitations, as detailed below.

However, these should be fairly easy to sort out.

Furthermore, I think the manuscript presentation could itself be made even better in one or two places, as detailed below

ABSTRACT

Generally clear.

However, we need a brief message regarding how large a change would be required to move to either the feasible approach (FA) or optimal diets (OD). 

It would be helpful even if you just said " the OD would typically involve doubling the intake of healthy foods, and halving the intake of unhealthy foods" .

(And that might be reasonably close to many of the actual detailed values chosen).

INTRODUCTION

Generally fine, 

and commendably brief.

METHODS

The Life Table methodology is uncontroversial.

The key issues with such models tend to be the choice of effect sizes, lag times and sensitivity analyses.

But firstly, a simple presentational issue.

Page 5, para beginning "Diets vary between individuals and settings.."

This para is very difficult to read.

I would suggest a modest revision to juxtapose the three levels for each food group, thus:

" The optimized diet (OD) values were set where dose-response data on consumption indicated no additional mortality gain in further increasing or decreasing intake (i.e., the impact on mortality plateaued). As a compromise between the typical diet and the optimal diet, we also considered a feasibility-approach diet (FA), which was chosen as the mid-point for each food group between the typical diet and OD. "

Then:

" In each case, dietary intake was improved from the typical western diet through feasible to optimal levels - 

 whole grains (fresh weight): 50g, FD 125g and OD 225g 

 vegetables, 250g, 325g and 400g , respectively"

 and so on.

Thus, at a glance, readers can then see how modest or radical were each of the required changes.

Page 5.

The extensive sensitivity analyses are to be commended.

Likewise using NutriGrade to assess the quality of the published evidence

The https://food4healthylife.org/ website is ambitious, but impressive.

The para on time lags is just plain wrong. 

The authors currently state:

"As health gains from diet changes are generally linked to reduction in cardiovascular disease

and cancers,[2-5] it is likely that effects of changes will increase gradually over a few

decades. We therefore assumed delayed onset of benefits and that time to full effect were 30

years from diet change with a gradual, linear increase in effect (e.g., the effect was 20% after

6 years), and conducted sensitivity analyses with 10 years and 50 years delays until full effect."

In fact, many studies have reported event and mortality changes starting within months of a change in the population, and becoming substantial within a few years (not decades). 

That needs to be factored in, and the modelling estimates then need to be repeated. 

The good news is that the estimated benefits could happen much sooner (see my comments below on Page 7, Para 2).

The current para on lag-times therefore needs to be deleted, and a more up-to-date summary inserted, referencing the relevant literature. 

That might include, for instance, mention Lancet, 2011; 378: 752-753, and European Heart Journal (2011) 32, 1187-1189, Can dietary changes rapidly decrease cardiovascular mortality rates? doi:10.1093/eurheartj/ehr049. The subsequent publications which cite these early scoping reviews might also be worth a quick look.

I would consider that this minor revision of the model would be essential before being accepted for publication.

RESULTS

Generally good.

From a publishing tactical point of view, you might enjoy MUCH higher citation rates if you detail the USA results in the Results text, 

and place the European & Chinese results in the Appendix.

In terms of face validity, one might have expected bigger benefits from increasing veg compared with fruit,

And smaller benefits from reducing red meat, compared with processed meats.

Page 7 Para 2 currently states

"Conversely, decreasing time-to-full-effect from 30 years to 10 years, health gains for 20-yearold

females and males increased by almost a year (i.e., by 3-7%), whereas gains increased by

three to four years for 60-year-olds (i.e., 53-78%) and more than two years for 80-year-olds

(i.e., 250-267%)."

I suspect that these values may be close to the ones you obtain after you revise the lag-times in your model, as advised above.

DISCUSSION

Generally good.

Making the Food4HealthyLife calculator freely available online is commendable.

However, the language needs attention.

These outputs are estimates, based on an early version of a new model, using imperfect data, and awaiting replication. 

They do not "show" anything. The appropriate language should therefore be more cautious, using words such as "suggest", "indicate" etc. 

The definitive language therefore needs to be toned down. Eg "Our results show…."

The Concluding paragraph is thus much better, mainly saying "could".

Page 8. The para begging " Our method has several strengths and limitations " is messy.

It would be best to split the messages into a short para on strengths, and a longer one on Limitations, adding the further limitations that I and the other referees have identified.

None are catastrophic, but it is tactically sensible to acknowledge each one here. That will pre-empt embarrassing letters to the journal, or more public criticisms. 

LIMITATIONS

Please add the further limitations that I and the other referees have identified.

These include:

You need to add a sentence or two critiquing GBD. While stupendous in scale, it has many imperfections. A five minute web-search will identify a few. 

The analysis appears to downplay olive oil, which proved very powerful in the large PrediMED RCT.

The authors could easily test a bigger effect in a quick sensitivity analysis.

Likewise, the OD for nuts should probably be 40g, rather than 25g, (Estruch NEJM 2028)

This model is at an early stage of development. 

Iterative improvements are likely to mean that the precise numerical outputs are therefore change; however, the main messages are likely to prove robust.

FIGURES

Fig 1 is good

Fig 2 is visually outstanding, while demonstrating some complex effects in a very accessible way. 

*** Reviewer #2: 

That nutrition plays a role in the pathogenesis of many chronic diseases is well known. It is also known that manipulation of specific molecular pathways by nutritional (dietary restriction) and exercise training can impact aging and life expectancy. However, it seems that energy intake/balance (and adiposity) plays a major role in determining this relationship, with macronutrient composition potentially playing a contributing effect. 

The findings of this paper are potentially interesting but it is very difficult to understand the assumptions and mathematical modelling supporting the data. Johansson et al. (2020) framework (ref.12) for measuring life expectancy from disease onset for specific conditions is not very clear, and less so the use of "changes in single dietary components" to determine life expectancy gains in this paper. The authors must make a much better job in explaining how they have calculated life expectancy gain for each single food (or combinations of foods) based on data from meta-analyses (providing dose-response data on the impact of various food groups on mortality). Again, there are a lot of assumptions based on results from meta-analysis data from epidemiological studies that by definition cannot demonstrate causality but just associations. Even more problematic are the assumptions and estimates of life expectancy gain based on duration of change. The time perspective of diet change adds another layer of uncertainty. 

The authors stated that the background data have been adjusted for factors such as smoking, exercise, age, and sex. Did they adjusted for BMI, or most importantly weight gain since early adulthood. 

*** Reviewer #3: 

Alex McConnachie, Statistical Review

The paper by Fadnes et al looks at modelling the possible impact of sustained dietary changes from a western-style diet towards a more optimal diet, on life expectancy. This review considers the statistical elements of the paper.

On the one hand, this could be a very short review, since there are no statistical methods used in the paper, in the usual sense of hypothesis testing and statistical model fitting. Nevertheless, I read the paper with interest, and I have a few comments that I hope will improve the paper.

The descriptions of a typical western diet, the optimal diet, and the feasibility-approach diet were nice and clear. The idea of the FA diet lying half-way between the typical and OD was a good one, as an attempt to reflect what might be practically achievable by most people, though I thought some of the calculations looked wrong. For example, if the typical intake of nuts is taken as zero, and is 25g for the OD, should it be 12.5g (not 25g) for FA? For fish, should it be 125g (not 100g) for fish (midway between 50 and 200g)? The values for the FA diet do not seem to match the way it was described.

The simulation approach looks OK, but was only repeated 200 times for each scenario. Is this enough? Whenever I use a sampling-based method, I tend to use thousands of replications - nowadays, computing power is not an issue with these things, so I think it is worth erring on the high side.

The tables and figures were a little confusing. Looking at Table 1, at the estimates for a 20-year-old, and at Figure 1, I would expect the estimates and uncertainty intervals to be the same - as far as I can tell, they are presenting the same data. But they are not - e.g. for female, for a change in legumes from 0 to 200, the table gives values of 2.0 (1.0, 3.2), but the figure reports 2.00 (0.90, 3.10). Most of the values that I checked differ slightly between the table and the figure. Also, in the combined document I was given to review, there is another version of Figure 1 with different estimates and intervals from both table 2 and the first version of figure 1. These inconsistencies are worrying.

In fact, since the figure shows the same data as the table (or should, as far as I can tell) I would suggest the table is redundant - the figure is more visually appealing, and includes the actual estimates, so is preferable. The tabular format may be better for the supplement, in order to pack in more information per page, but that doesn't really matter.

Another comment on the values reported in the figures - why do all the estimates have zero in the second decimal place? This is too unlikely to be plausible.

In the figures, the uncertainty intervals should not be referred to as "CI" - they are not confidence intervals. "UI" would be more appropriate. Also, is "Effect" the right word for the estimated life years gained?

In terms of layout, the figures have the TW->FA and TW->OD estimates for each food group together. This is fine, but visually, it would help if each pair of estimates were separated slightly. Alternatively, you could put all the TW->FA estimates together, followed by all the TW->OD estimates, with a gap between the two sections. I guess there are lots of ways these figures could be modified, and finding the optimal layout is not easy.

Figure 2 in the paper looks great, except for the fact that it is very hard to tell some of the colours apart. I don't know what could be done about this.

***

[LINK]

---

## [Decision Letter · Decision Letter 2]

5 Dec 2021

Dear Dr. Fadnes,

Thank you very much for re-submitting your manuscript "Estimating impact of food choices on life expectancy: A modeling study" (PMEDICINE-D-21-04011R2) for consideration at PLOS Medicine.

I have discussed the paper with editorial colleagues and it was also seen again by three reviewers. I am pleased to tell you that, provided the remaining editorial and production issues are fully dealt with, we expect to be able to accept the paper for publication in the journal.

[LINK]

Please let me know if you have any questions, and we look forward to receiving the revised manuscript.   

Sincerely,

Richard Turner, PhD

rturner@plos.org

Requests from Editors:

At the start of the abstract, please consider adapting the text to "Interpreting and utilising the findings of nutritional research can be challenging ...", or similar. 

Early in the abstract, please adapt the tense to "... we used life-table methodology ...".

In the abstract, please specify the nature of the uncertainty interval quoted at first use, e.g., "... the United States (10.7 [95% Uncertainty Interval 5.9-14.1] years) ...".

Generally, square brackets should be used within parentheses throughout. 

In the abstract and throughout the text, please use the format "age 20 years ...".

In the abstract, please add spaces to "... females: 2.0 ..." and similar.

In the abstract, should that be "... 20-year old women"?

Please adapt the current final sentence of the "Methods and findings" subsection of the abstract to "Using NutriGrade the overall quality of evidence was assessed as moderate.", or similar, and move this before the final sentence of the subsection.

Please trim the discussion of limitations in the abstract to one sentence to conclude the "Methods and findings" subsection. 

In the author summary, under the subsection "What did the researchers do and find?" please either add a new initial point to briefly describe the approach used or, alternatively, the existing first point could be adapted to do this. 

In the Methods section (main text), please state when the Pubmed search was done. 

Noting the long paragraph on limitations in the discussion, please break this up into at least two paragraphs. 

Please remove footnotes (this information can be integrated into the text). 

Please use an initial capital for "Western" (diet) consistently, throughout. 

Throughout the text, please remove spaces from the reference call-outs, e.g., "... data sources [28,29]." in the Discussion. 

In the reference list, please abbreviate journal names consistently (noting reference 31 and others). 

Please spell out the institutional author names for references 1 & 17. 

For reference 3 and any other relevant references, please adapt the journal name abbreviation to "BMJ".

Please remove "[cited 2020]" from reference 12. 

Please use the journal name abbreviation "PLoS ONE" for reference 13. 

Comments from Reviewers:

*** Reviewer #1: 

Thank you for taking on board the various reviewers' suggestions, which were all intended to be constructive.

I think you have now made your good paper even better.

*** Reviewer #2: 

The authors have constructively responded and addressed my concerns.

*** Reviewer #3: 

Alex McConnachie, Statistical Review

I thank the authors for their responses, all of which are satisfactory. I have no further comments.

Regarding the figures, I find R is highly flexible.

***

[LINK]

---

## [Editor Report · Decision Letter 3]

11 Dec 2021

Dear Dr Fadnes, 

On behalf of my colleagues and the Academic Editor, Dr Fontana, I am pleased to inform you that we have agreed to publish your manuscript "Estimating impact of food choices on life expectancy: A modeling study" (PMEDICINE-D-21-04011R3) in PLOS Medicine.

Prior to final acceptance, please address some minor points:

The correct name of the GBD study is used, but we suggest writing "Global Burden of Disease" (rather than "Diseases") where the short name is used, e.g. early in the abstract. 

At first mention of "Food4HealthyLife" in the abstract, please amend the text to "... calculator that we provide online could be useful ..." or similar.

Where "TRIPOD" is referred to late in the Methods section, please remove "tried to" (e.g., "We adhered to the ...").

Please add "95%" to "Uncertainty Interval", if appropriate. 

Early in the Results section (main text) please add "Uncertainty Interval" to the first such quoted. 

PRESS

Sincerely, 

Richard Turner, PhD 

rturner@plos.org